# AT Homopolymer Strings in *Salmonella enterica* Subspecies I Contribute to Speciation and Serovar Diversity

**DOI:** 10.3390/microorganisms9102075

**Published:** 2021-10-01

**Authors:** Jean Guard, Adam R. Rivers, Justin N. Vaughn, Michael J. Rothrock, Jr., Adelumola Oladeinde, Devendra H. Shah

**Affiliations:** 1US Department of Agriculture, US National Poultry Research Center, Athens, GA 30605, USA; Justin.vaughn@usda.gov (J.N.V.); Michael.rothrock@usda.gov (M.J.R.J.); ade.oladeinde@usda.gov (A.O.); 2US Department of Agriculture, Genomics and Bioinformatics Research, Gainesville, FL 32608, USA; Adam.rivers@usda.gov; 3Department of Veterinary Microbiology and Pathology, Washington State University, Pullman, WA 99164, USA; dshah@wsu.edu

**Keywords:** *Salmonella enterica*, food safety, genome, theory, single nucleotide polymorphisms, recombination, serovar

## Abstract

Adenine and thymine homopolymer strings of at least 8 nucleotides (AT 8+mers) were characterized in *Salmonella enterica* subspecies I. The motif differed between other taxonomic classes but not between *Salmonella enterica* serovars. The motif in plasmids was possibly associated with serovar. Approximately 12.3% of the *S. enterica* motif loci had mutations. Mutability of AT 8+mers suggests that genomes undergo frequent repair to maintain optimal gene content, and that the motif facilitates self-recognition; in addition, serovar diversity is associated with plasmid content. A theory that genome regeneration accounts for both persistence of predominant *Salmonella* serovars and serovar diversity provides a new framework for investigating root causes of foodborne illness.

## 1. Introduction

Approximately 30 of 1500 *Salmonella enterica* subspecies I (*S. enterica*) serovars are persistent agents of foodborne illness in people [1]. Despite improved biosecurity throughout the food production pipeline, reduction of salmonellosis has plateaued over 20 years [2]. The inability to reduce salmonellosis indicates new approaches to understanding the biology of this important pathogen are needed. Recently, the most commonly occurring single nucleotide polymorphism (SNP) that caused disruption of a gene in *S. enterica* serovar Enteritidis (Enteritidis) was identified, and it was deletion of a single adenine in a homopolymer string of 8 nucleotides (nt) within the fimbrial gene *sef*D [3]. Mutational analysis, phenotype microarray, and infection experiments in the egg-laying hen indicated that the *sefD* mutation increased organ invasion and mortality in hens, disturbed egg production, enhanced growth of the pathogen to high cell density, and otherwise behaved as a regulator of dimorphism of phenotype [4]. As a result of this finding the performance of a killed vaccine for hens was enhanced by increasing SefD in preparations [5]. The drastic change in biological phenotype imparted by the single base pair deletion suggested that characterization of purine homopolymer strings of adenine, AAAAAAAA, and its pyrimidine base pair (bp) of thymine, TTTTTTTT, in *S. enterica* should be explored. It is not evident that *S. enterica* of different serovars would have conserved AT 8+mer content, because it has a mosaic genome with frequent inversions of sections of the core genome as well as differences originating from mobile genetic elements such as bacteriophage, transposons, insertion elements, and plasmids.

Previous analysis of homopolymeric A and T strings across 81 bacterial and 18 archeal genomes showed that the motif in bacteria was hypermutable, occurred preferentially at the 5′ end of genes, and was not biased for AA- or TT- encoded amino acids [6]. Furthermore, detailed analysis of these tracts in the *inlA* gene of *Listeria monocytogenes* supported the premise that AT homopolymeric tracts were part of a robust regulatory mechanism that facilitated a reversible, rapid evolutionary mechanism facilitating adaptation and phase variation in gene regulation. Finding that *S. enterica* serovar Enteritidis, the world’s leading cause of foodborne salmonellosis, had the hypermutable gene *sefD* linked to phenotypic phase variation suggested that review of the *S. enterica* genome for AT homopolymeric tracts was warranted.

In addition to the research already cited, there is other evidence that AT homopolymer tracts, referred to in this manuscript AT 8+mers, constitute an important regulatory element in bacteria. It is a DNA motif suggested by conformational studies to bend DNA out of the Z-conformation [7]. Polyadenine regions can impact gene regulation in prokaryotes and can contribute to microsatellite instability in eukaryotes [8,9,10,11]. It has been shown that homopolymer nucleotide strings contribute to non-programmed slipped strand replication and the accumulation of errors in DNA [12,13,14]. Thus, the physicochemical impact of these strings was another reason to catalogue this motif in the genome of *S. enterica.*

The incidence of homopolymer strings in completed genomes of subspecies I of *S. enterica* and two different taxonomic classes, namely Gammaproteobacteria and Bacilli, were compared to better understand if *S. enterica* was at all unique. Reference genomes of *S. enterica* serovars Enteritidis and Typhimurium were analyzed to locate mutated AT 8+mers in genomes because they are from different genomic lineages and have been extensively sequenced and annotated. Together they cause approximately 40% of all foodborne salmonellosis in the US [1]. *S. enterica* serovar Gallinarum was included in the same analysis because it is a biological outlier within subspecies I that does not cause foodborne disease. Unlike either Enteritidis or Typhimurium serovars, Gallinarum causes disease in poultry resulting in high morbidity, mortality, and economic loss. A comparative approach is useful for linking phenotype to single nucleotide polymorphisms occurring in genomes [15]. In this study, the three genomes were compared to better understand the association the motif might have with naturally occurring mutation that disrupts open reading frames of genes.

### Additional Background on Poultry-Associated S. enterica Serovars

*S. enterica* serovars Enteritidis and Typhimurium differ biologically [16]. Epidemiological patterns for the two predominant pathogens also differ. Enteritidis is an exceptional *Salmonella* pathogen in part because it efficiently contaminates the internal contents of eggs produced by otherwise healthy-appearing hens. It produces a high molecular mass (HMM) O-antigen, which not only protects killing of the pathogen by the host but also acts as a protective capsule in the hostile environment of the egg [17,18,19]. Typhimurium is also resistant to complement but it does not produce HMM O-antigen, and thus it does not survive in the internal contents of eggs to an extent that can be detected by epidemiological surveillance. Both Typhimurium and Enteritidis can contaminate a broad spectrum of other food sources such as poultry carcasses, other meats, and fresh vegetables. Both serovars can invade organs and survive, which contributes to systemic spread during infection [20]. Variation between strains within each serovar occurs but serovar characteristics and general genome organization are maintained [21,22]. There are serovar-specific patterns in plasmid carriage and fimbrial genes. Comprehensive reviews are available on the similarities and differences between *Salmonella* serovars [23,24,25,26,27].

*S. enterica* serovars Gallinarum and Enteritidis are genetically closely related [28]. Serovar Gallinarum’s antigenic formula is 1,9,12:-:-, which indicates it has the same lipopolysaccharide O-antigen epitopes as Enteritidis; however, it lacks both H1 and H2 flagellin proteins and is thus non-motile. Both Gallinarum and Enteritidis can contaminate the internal contents of eggs; however, Gallinarum has mutations and rearrangements that restrict its host range to the avian host, possibly by reducing immunological response to infection and thereby facilitating systemic infection [20]. Thus, the most striking differences between the foodborne pathogen Enteritidis and host-restricted Gallinarum is that the latter makes poultry sick, reduces egg production, and causes mortality. In contrast, hens infected with Enteritidis often appear healthy, remain in production, eggs become contaminated internally, and are a source of foodborne illness. The ability of Enteritidis to spread through flocks that appear healthy was one of the contributing factors in its world-wide spread through the layer industry. The differences in the epidemiology, association with food, and virulence characteristics of the three pathogens, all of which occur in the poultry environment, supported comparing them to better understand the association between the AT 8+mer motif and naturally occurring mutation of an important food borne pathogen. Other pathogenic Salmonellae and bacteria with different taxonomy were also included in analysis to facilitate comparison to results obtained by others [6].

## 2. Materials and Methods

### 2.1. Salmonella enterica Subspecies I Strains Analyzed for Strings of Homopolymers

A database of 49 completed genomes of *S. enterica* subspecies I (taxid:59201 was accessed at the National Center for Biotechnology Information (NCBI) [29] (Appendix A), and the last accession date was 20 August 2021. *S. enterica* serovars Typhimurium, Typhi, and Enteritidis genomes were over-represented compared to other serovars, and together, they comprised 39.4% of all completed genomes available at the time of analysis. Only 51.2% of *S. enterica* subspecies I genomes had a complete adenylate cyclase (*cyaA*) gene, which is required for virulence as a foodborne pathogen; thus, analysis was restricted to completed genomes to avoid issues associated with incomplete assembly and annotation. The other sequences were plasmids, which were also evaluated for the AT 8+mer motif. Genome CP018657 was excluded from analyses due to errors. Genome databases at NCBI show homopolymer strings, as well as other combinations of low-complexity regions, in lower-case gray font, because there is recognition that some sequence strings might be susceptible to alignment error; thus, they required masking during the alignment process. For the BLAST searches conducted here, each gene was observed for high fidelity of surrounding regions; therefore, it is unlikely low complexity impacted observed alignments. For *S. enterica* subspecies I, 12 complete genomes were assessed for Typhimurium, Enteritidis, and Typhi, and another grouping included a mixture of 12 serovars often recovered from foods. It is important to note that the NCBI is in the process of consolidating redundancy in over-represented genomes; thus, the total number of genomes represented by high quality and well-annotated accession numbers may be larger than it appears as redundant material is condensed and re-annotated (accessed on 26 August 2021; https://www.ncbi.nlm.nih.gov/refseq/about/prokaryotes/reannotation/).

### 2.2. Other Bacteria Analyzed for Comparison to S. enterica

To compare homopolymer strings in *S. enterica* subspecies I, AT and GC 8+mer homopolymers were tabulated from 42 strains encompassing 5 other genera in Phyla Proteobacteria (*Escherichia coli, Proteus mirabilis, Shigella sonnei, Yersinia pseudotuberculosis, Vibrio vulnificus* (chromosome I and II))*,* and 5 genera of Firmicutes *(Staphylococcus aureus, Streptococcus pyogenes, Enterococcus faecalis, Bacillus anthracis and Bacillus cereus)* (Appendix A). *Salmonella, Shigella,* and *Escherichia* are all members of the Family Enterobacteriaceae, whereas *Staphylococcus, Streptococcus, Enterococcus*, and *Bacillus* are in the Family Bacilli. At least 3 complete genomes of the other genera were assessed, including 12 genomes from *Escherichia coli* (Table 1).

### 2.3. Statistical Analysis of Kmer Content

Strings of homopolymers of different lengths were entered as text in the find function and the genomes were searched. Searches would count single locations multiple times if the length exceeded 8nt, e.g., kmers of 10nt would often be counted 3 times as 8, 9, and 10nts. Thus, each kmer was reviewed for length to make sure a single location was counted only once, and then it was catalogued according to total length. Review for undercounting homopolymer strings due to wrapping at end of lines in sequence reports was also done; thus, the Geneious or NCBI search programs were preferred. The genomes of interest were stored in SeqBuilder Pro, Lasergene V16.0.0 352 (DNASTAR, Madison, WI, USA) and in Geneious V2030.0.3 (Biomatters, Inc., San Diego, CA, USA) format (https://www.geneious.com, accessed on 15 July 2021). Results were copied into an Excel “.csv” file as Unicode text Microsoft Excel for Mac, V16.16.20 (200307). The text to column feature, and appropriate delimiters, were used to produce columns of data to calculate distance between nucleotide strings. The average, standard deviation, and median values between AT 8+mer homopolymers were then calculated. Homopolymer strings of all 4 nucleotides, ranging from 5–20 nucleotides, were counted in *S. enterica* serovars and other genera and results are shown in columns F and G of Table 1. To account for every AT kmer of at least 8 nucleotides, the longer motifs were added to 8mers in further analyses; thus, the term AT 8+mer is applied throughout to describe the motif. Ttest analysis was used to determine if differences in kmer counts between groupings were significant at *p* < 0.01.

*S. enterica* subspecies I serovar Typhimurium LT2 was the reference genome to produce a common denominator to normalize genomes of different sizes (Table 1). Thus, values greater than 1 indicated that more than the expected number of motifs were observed in comparison to *S. enterica* after normalizing for the size of the genome, and less than 1 indicated fewer motifs were observed.

AT 8+kmers for Typhimurium LT2 were classified as intergenic, intragenic, or regulatory using Genious Prime 2020.0.3 (Appendix A). Another approach used to establish a baseline incidence of AT 8+mers occurring in genes was to generate a list of random numbers using the 4600 predicted genes of the reference genome. Two hundred random numbers were generated between 1 to 4600 corresponding to numbered genes, a FASTA file was then compiled, and the number of AT 8+mers within the randomly generated sets was determined.

### 2.4. Locating AT 8+Mers within Mutated Genes

*S. enterica* serovar Typhimurium LT2 (NC_003197.2) was used as the primary reference sequence to name genes and gene functions [30]. The two other reference genomes were Enteritidis strain P125109 and Gallinarum strain 9184, with respective NCBI accession numbers of NC_011294.1 and CP019035.1 [31,32]. Classes and functions of proteins were assessed at https://uniprot.org (accessed on 20 August 2021). Determining impact on open reading frames (ORFs) within annotated genes was done with Genious V2030.0.3. In addition, online software available at https://web.expasy.org (accessed on 26 August 2021) was used to translate proteins and align amino acids with nucleotides. BLAST analyses and generation of reverse complemented sequences used the NCBI website (https://blast.ncbi.nlm.nih.gov/Blast.cgi) (accessed on 26 August 2021).

## 3. Results

### 3.1. Homopolymer Strings of at Least 8 Nucleotides Were Dispersed in the Genome of S. enterica

The AT 8+mers were dispersed throughout the entire genome of serovar Typhimurium LT2 (Figure 1). The genome of reference strain *S. enterica* serovar Typhimurium LT2 is 52.2% GC. When data were expressed as ratios of AT:GC homopolymer strings, the AT 8mer homopolymers (e.g., AAAAAAAA and TTTTTTTT) were much more prevalent than GC 8mers in the reference genome (Figure 2). In total, there were 294 AT 8mers and 11 GC 8mers in the reference serovar, which is a ratio of 27 AT 8mers to every GC 8mer. AT strings longer than 8bp were less frequently observed (Figure 2). On average, the motif occurred every 16,450nt (Appendix A). The range of AT 8+kmer distance was 11 to 117,141nt, and the median was 11,498nt (Appendix A). Distances of 51,975nt or greater between motifs were over 3 standard deviations and were thus possibly deficient in AT 8+mers. Of 13 putatively deficient regions, the 4 longest regions were assessed for phage genes, pseudo genes, insertion elements, transposases, ribosome binding sites, and regulons. The 4 regions were located between nucleotides (i) 1368633–1444823 (76,198nt), (ii) 2612956–2730097 (117,148nt), (iii) 4124625–4209022 (84,404nt), and (iv) 4342879–4418289 (75,418). At this time, no feature could be found that differentiated AT 8+mer deficient regions from regions separated by a shorter distance.

### 3.2. The AT 8+mer Motif in Bacteria Was Specific to Genus and Species

Results of comparisons between Bacteria are shown in Table 1. Details include: (1) AT 8+mers in *S. enterica* groups were significantly more frequent than what was observed for *E. coli* (*p* < 0.005); (2) across bacterial classes, *Vibrio vulnificus* cII had a minimum of 90.0 AT 8+mers and *Proteus mirabilis* had a 712.7; (3) standard deviations between strains in each Genera ranged from 2.3 for *Yersinia pseudotuberculosis* to 84.1 for *Enterococcus faecalis*; (4) all the genera examined, including those that differed by Phylum and Class, had a relative paucity of GC 8+mers as compared to AT 8+mers; thus, it appears there is a bias for Bacteria maintaining AT 8+mers in genomes, or inversely, selecting against GC 8+mers; (5) each species appeared distinctly different from others; (6) *Vibrio vulnificus* had 180 and 90 AT 8+mers in chromosomes cI and cII, respectively; thus, AT 8+mer content might be a chromosomal characteristic that maintains the separation of the two chromosomes.

Results based upon taxonomic class were an average of 379.57 and 430.8 AT 8+mers, respectively, for the Gammaproteobacteria and Bacilli. In addition, standard deviations were large for both classes, namely 174.51 and 251.24 respectively. The range reflected by standard deviations masks the individuality of each genus and species. For example, *Bacillus anthracis* had an average of 432 AT 8+mers, whereas *Bacillus cereus* had an average of 700.3. Results from *S. enterica* subspecies I had a relatively small overall standard deviation of 12.83 AT 8+mers across 24 serovars, which agrees with the classification of *S. enterica* subspecies I as a single genomic grouping.

### 3.3. The AT 8+mer Motif in the Chromosome Is Not Specific to Serovar

As mentioned in the previous section, AT 8+mers per *S. enterica* grouping was from 315.6 to 332.6, and the average was 322.2 +/− 12.83 AT 8+mers (Table 1). The length of the homopolymer is proposed to impact the physicochemical bending properties of DNA; thus, we wanted to account for every kmer of 8 nucleotides or more. Results from analysis of AT 8+mers between *S. enterica* serovars were: (i) The incidence of AT 8+mers in the reference genome for serovar Typhimurium LT2 was the lowest of the 12 strains in the group, which suggests that using the serovar as a reference would not over-estimate the incidence of AT 8+mers for *S. enterica* or other genera; (ii) the standard deviations for AT 8+mers in serovar Typhimurium and in the group of mixed serovars were, respectively, 13.0 and 13.9; (iii) serovars Enteritidis and Typhi, with respective standard deviations of 10.5 and 5.9, appeared more clonal than Typhimurium, and this finding agrees with current knowledge; (iv) the foodborne serovars, namely Typhimurium, Enteritidis, and the group of mixed serovars, had a more variable motif content than human-restricted Typhi. Overall, the *S. enterica* serovar groups were not significantly different from each other although Typhi had the lowest standard deviation. There were not enough completed genomes of the poultry-restricted serovar Gallinarum to include it for this analysis; however, a reference genome of this important host-restricted poultry pathogen is evaluated in the following text.

### 3.4. Characterization of the AT 8+mer Motif in Poultry-Associated Serovars of Salmonella

The association of motif content with gene function was catalogued. Appendix A lists 294 intergenic, regulatory, and gene AT 8+mer loci in serovar Typhimurium LT2; of these, 131 were intergenic, 150 were in genes, and 13 were in regulatory regions. Appendix A lists 185 genes and regulatory regions with AT 8+mers, which were located by comparing reference genomes of serovars Typhimurium, Enteritidis, and Gallinarum; thus, 20 more genes and regulatory regions were identified using the comparative approach. Gene functions were assigned in the last column. Some genes in serovars Enteritidis and/or Gallinarum did not have homologs in the Typhimurium reference strain, and vice versa. Serovars Typhimurium, Gallinarum, and Enteritidis each had 3, 22, and 5 pseudogenes with AT 8+mers, respectively, while each genome had a total of 40, 287, and 96 pseudogenes. Thus, respectively, 7.5%, 8.4%, and 5.2% of pseudogenes involved the motif.

The regulons with the motif encompassed mostly metabolic functions. Regulons associated with the genes STM2277, STM0664, and STM4585 have some association with antibiotics (references not shown). Another class of gene that could be of interest for targeting of antimicrobials include transporters, and there are 20 listed in Appendix A [33]. Cell surface molecules that include genes with the motif include colonic acid, lipopolysaccharide, flagella, and fimbria. A virulence factor with the motif is MviM.

### 3.5. Location of AT 8+mers within Mutated Open Reading Frames

Table 2 lists 22 genes that are a subset of those from Appendix A. Serovar Gallinarum had 17 pseudogenes that were intact in serovars Enteritidis and Typhimurium. There were 4 pseudogenes in both serovars Enteritidis and Gallinarum that were intact in Typhimurium. STM1666 was unusual because it was a pseudogene in serovar Typhimurium, absent in Gallinarum, and intact in Enteritidis. STM1666 is thus one of the few genes that differentiates serovar Enteritidis from both Gallinarum and Typhimurium. The gene was present in 31 other Salmonellae genomes available at NCBI, but no serovar was reported for these and no other information was present. There was no homolog of STM1666 in other bacteria.

The location of the first adenine or thymine in homopolymer strings in the subset of mutated genes was located relative to start codons ATG or GTG. Results were that 8, 7, 6, and 1 gene(s) fell into the 1st, 2nd, 3rd, and 4th quartile of gene lengths. These findings thus agree in concept with others that homopolymer tracts are found closer to the 5′ end of open reading frames [6]. However, we suggest that these data are more accurately summarized as AT 8+mers of *S. enterica* do not often occur near the 3′ end of genes. An area of future research is to determine if taxonomic classes of bacteria differ in location of mutable homopolymer strings within open reading frames.

### 3.6. S. enterica Plasmids Have AT 8+mer Motifs Possibly Associated with Serovar Preference

Some of the most important foodborne *Salmonella* serovars harbor a large low-copy virulence plasmid, which is consistently present even when plasmid profiles differ between strains [34,35]. *S. enterica* serovar Typhimurium, and related immunogenic variant with LPS O-antigens [1,4,5,12] has one of the largest virulence plasmids (pSLT) (NC_003277.2) [30]. At 93,933nt, pSLT would be expected to have 5 or 6 AT 8+mers if the average of motifs found within the chromosome, which was 16,450nt, applied to the extrachromosomal plasmid. Typhimurium pSLT had 6 AT 8+mer loci (Table 3). No evidence of hypermutability of these loci was found within serovar by BLAST analysis; thus, AT 8+mers within serovar appeared conserved. The motif in plasmids of the other two serovars differed substantially from each other and from Typhimurium (Table 3). The serovar Enteritidis plasmid pSENV (CP063701.1) and Gallinarum SG9 strain 9 plasmid (CM001154.1) were 59,372 and 87,371 nucleotides, respectively, and each had 4 AT 8+mers. Overall, there was substantial variation between the AT 8+mer motif in each of the plasmids for the three serovars reviewed, and substantial serovar-related differences in plasmid profiles have been reported [36]. While these results suggest that at least the pSLT virulence plasmids have serovar preference, there are not enough plasmids sequenced across subspecies I to comment further. Analysis of the relationship between AT 8+mer content of plasmids and serovar is a topic for future investigation.

## 4. Discussion

The AT 8+mer motif was shown here to be associated with (1) *S. enterica* speciation, (2) serovar designation via details of the AT 8+mer in plasmids, (3) hypermutability in genes and regulatory regions that impact phenotype, growth potential, virulence and metabolism of *Salmonella enterica* subspecies I. AT 8+mers influence microbial biology. For example, A and T homopolymers impact transcription termination in Archea [37]. The canine herpesvirus thymidine kinase gene has mutational hotspots at stretches of 8 adenines [38]. T7 bacteriophage RNA polymerases undergo transcription slippage at A and T homopolymers [39]. As mentioned previously for *S. enterica* serovar Enteritidis, a mutational hotspot in 1 of 8 adenines increased virulence [3]. We did not conclude that mutation in the motif was primarily in the 5′ region of genes; however, it is a subject that requires development of better algorithms to address in detail [6]. These results suggest that the AT 8+mer motif facilitates maintenance of *S. enterica* as a persistent and important foodborne pathogen, as well as impacting epidemiological outcomes by contributing to ecological adaptation [40].

If AT 8+mers are mutational hotspots, then there must also exist a mechanism for repair. Otherwise, evolution of any one serovar of *S. enterica* would be unidirectional towards extinction. There are several examples of *Salmonella* serovars, e.g., Typhimurium, Enteritidis, Newport, Infantis, and Heidelberg, that persist over decades; however, the majority of the 1500 serovars within subspecies I cause illness inconsistently, rarely, or never [1,16]. For this reason, we theorize there is another function for AT 8+mers. It is proposed that chromosomal AT 8+mers align sections of genomes during replication and DNA repair processes. This function would result in repair of mutations occurring between stretches of wildtype AT 8+mers [41,42]. It would also account for an inherent mechanism of self-recognition, which would facilitate preferential, but not exclusive, DNA exchange within the species. The pan-genome of *S. enterica* subspecies 1 has a mosaic structure, with frequent inversions, deletions, and insertions occurring between serovars; however, the chromosomal arrangement of many *Salmonella* lineages is comparatively stable [25,32,43,44]. Thus, the motif would account for (i) the stability of some serovars with conserved genome features that are persistent, e.g., serovar Typhimurium [1], (ii) the occasional emergence of a new serovar that happens to undergo clonal expansion in an environment favorable for growth, e.g., serovar Tennessee in peanut butter [45,46], (iii) the rare emergence of a hybrid strain following a major recombination event that results in rapid proliferation of a serovar with new biological properties, e.g., serovar Enteritidis [47], and (iv) the periodic emergence and disappearance of serovars that are not optimized for the survival in the environment in which they are generated. Another finding that the AT 8+mer impacts speciation is that the two chromosomes of *Vibrio vulnificus* differ substantially and would thus never be predicted to coalesce.

## 5. Conclusions

These results support and add to the concept that the homopolymer AT motif is an important regulator of bacterial phase transition, ecological persistence, and association with disease [6]. In regard to *Salmonella enterica* subspecies I, the motif is proposed to be important for maintaining optimal gene content, with the coincidental impact of contributing to serovar diversity [48,49,50,51]. Future research on the AT 8+mer contribution to these proposed functions will require proof of concept experimentation. Biological experimentation will focus on finding environmental niches that facilitate genomic exchange and repair mechanisms. Analyzing the impact of the motif on the safety of the food supply may require methods with detection limits that are orders of magnitude lower than those used to currently detect bacteria. This is because a successful recombinant may at first be a rare cell type [52,53]. Further analysis into the impact of AT 8+mers on the ability of *S. enterica* to survive and persist in environments associated with foodborne illness is thus warranted.

## Figures and Tables

**Figure 1 microorganisms-09-02075-f001:**
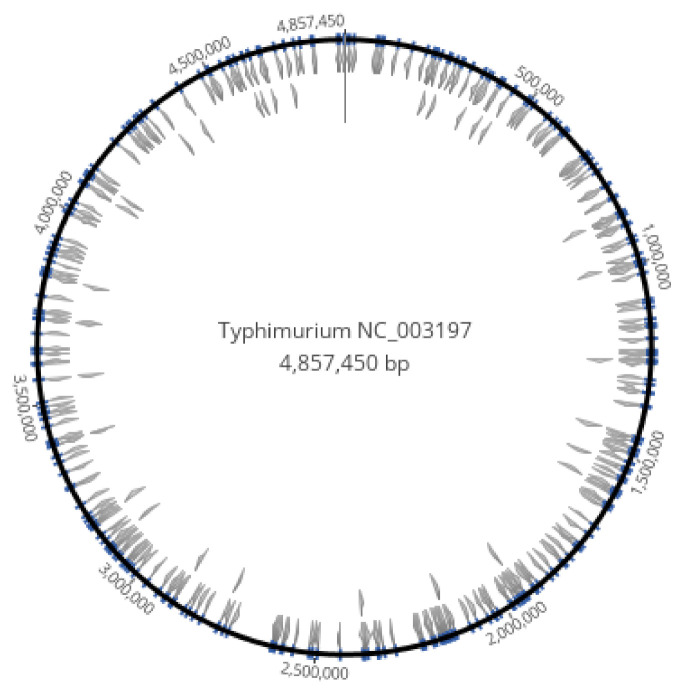
Locations of AT 8+mers in the genome of *Salmonella enterica* serovar Typhimurium LT2 NC_003197.2.

**Figure 2 microorganisms-09-02075-f002:**
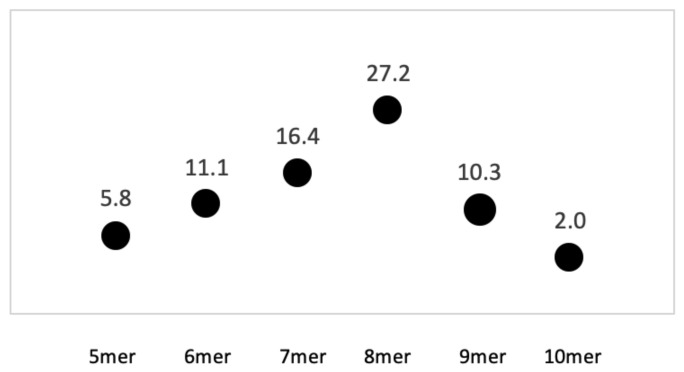
Ratios of AT homopolymers from 5 to 10 nucleotides in *Salmonella enterica* serovar Typhimurium LT2 NC_003197.2. The ratio of AT homopolymer kmers, either adenine or thymine but not mixed, to GC homopolymers was determined using Geneious software as described in text. The range in number of nucleotides per kmer searched was 5 to 10 (see legend label). Results show that a nucleotide motif of 8 was the most commonly encountered, and that approximately 27 AT homopolymers were found for every 1 GC AT homopolymer in the reference sequence of *S. enterica* LT2 NC_003197.2.

**Table 1 microorganisms-09-02075-t001:** Expected versus observed occurrence of homopolymer strings of 8 and more nucleotides in genomes of Bacteria.

Phylum	Genus Species ^1^	Other Genome Information	Number of Genomes Analyzed	Characteristics	Genome Size (bp)	Common Denominator (nt) ^2^	Expected Number of 8+kmers	Observed AT 8+mers	Observed GC 8+mers	Observed vs. Expecterd AT 8+mers ^3^	Observed vs. Expected GC 8+mers ^3^
Proteobacteria	*Salmonella enterica*	Typhimurium	12	Average	4,890,448	16,299	300.0	332.6	17.2	1.11	0.06
				stdev	50,356	---	3.1	13.0	3.6	---	---
Proteobacteria	*Salmonella enterica*	Enteritidis	12	Average	4,686,462	16,299	287.5	323.7	21.5	1.13	0.07
				stdev	20,384	---	1.3	10.5	4.2	---	---
Proteobacteria	*Salmonella enterica*	Typhi	12	Average	4,770,414	16,299	292.7	316.9	29.5	1.08	0.10
				stdev	60,270	---	3.7	5.9	3.2	---	---
Proteobacteria	*Salmonella enterica*	mixed	12	Average	4,713,701	16,299	289.2	315.6	17.2	1.09	0.06
				stdev	80,652	---	4.9	13.9	4.4	---	---
Proteobacteria	*Escherichia coli*	---	12	Average	5,087,133	16,299	312.1	281.9	18.3	0.90	0.06
				stdev	262,098	---	16.1	30.2	6.5	---	---
Proteobacteria	*Proteus mirabilis*	---	3	Average	4,124,431	16,299	253.0	712.7	15.7	2.82	0.06
				stdev	83,305	---	5.1	42.0	2.1	---	---
Proteobacteria	*Shigella sonnei*	---	3	Average	4,929,599	16,299	302.4	261.3	11.7	0.86	0.04
				stdev	90,607	---	5.6	8.4	3.5	---	---
Proteobacteria	*Yersinia pseudotuberculosis*	---	3	Average	4,802,245	16,299	294.6	429.3	120.3	1.46	0.41
				stdev	118,706	---	7.3	2.3	3.8	---	---
Proteobacteria	*Vibrio vulnificus*	chromosome I	3	Average	3,330,104	16,299	204.3	180.0	10.3	0.88	0.05
				stdev	79,423	---	4.9	14.0	4.9	---	---
Proteobacteria	*Vibrio vulnificus*	chromosome II	3	Average	1,756,668	16,299	107.8	90.0	3.3	0.83	0.03
				stdev	87,177	---	5.3	7.9	3.1	---	---
Firmicutes	*Staphylococcus aureus*	---	3	Average	2,948,373	16,299	180.9	108.3	0.0	0.60	0.00
				stdev	114,371	---	7.0	10.6	0.0	---	---
Firmicutes	*Streptococcus pyogenes*	---	3	Average	1,895,707	16,299	116.3	263.7	0.3	2.27	0.00
				stdev	42,370	---	2.6	15.4	0.6	---	---
Firmicutes	*Enterococcus faecalis*	---	3	Average	3,090,387	16,299	189.6	649.7	2.0	3.42	0.01
				stdev	117,259	---	7.2	84.1	3.5	---	---
Firmicutes	*Bacillus anthracis*	---	3	Average	5,228,732	16,299	320.8	432.0	1.3	1.35	0.00
				stdev	1349	---	0.1	11.5	0.6	---	---
Firmicutes	*Bacillus cereus*	---	3	Average	5,406,060	16,299	331.7	700.3	13.0	2.11	0.04
				stdev	16,615	---	1.0	53.7	6.1	---	---

^1^ Genomes included in analysis are listed in Appendix A with NCBI accession numbers. ^2^ The common denominater of 16,299 nucleotides (nt) used to normalize variation in geome size was obtained from *Salmonella enterica* subspecies I serotype Typhimurum LT2 (NC_003197.2) as described in text. ^3^ Values greater than one indicate more motifs observed than expected, less than one indicates fewer were observed than expected.

**Table 2 microorganisms-09-02075-t002:** AT 8+mer motifs associated with gene disruption or altered regulatory regions in *S. enterica* serovars Enteritidis, Table 1.

STM Gene Accession	SEEG Gene Accession	SEN Gene Accession	AT 8+mer Sequence	Common Name of Gene	Description of Target Gene	Gene Function	Biological Process
no homolog	SEEG9184_21515	SEN_RS22080	conserved (3 locations)	*sefC*	SEEG pseudogene	Has 3 AT 8mers in sequence; outer membrane fimbrial protein SefC, pseudo in SG, which has an extra A/T to make a 7mer	adhesion
STM0071	SEEG9184_20585	SEN_RS00360	conserved	*caiC*	SEEG pseudogene	crotonobetaine/carnitine-CoA ligase	transporter
STM0858	SEEG9184_16520	SEN_RS04155	conserved	unnamed	SEEG pseudogene	electron transfer flavoprotein-ubiquinone oxidoreductase	transporter
STM2020	SEEG9184_10295	SEN_RS10515	conserved	*cbiO*	SEEG pseudogene	cobalt transport atp-binding protein CbiO: B12 synthesis associated? Truncated, maybe shorter product?	transporter
STM2241	SEEG9184_09260	SEN_RS11570	conserved	*ssp*H2	SEEG pseudogene	E3 ubiquitin-protein ligase; induced by the SPI-2 regulatory ssrA/B	virulence factor
STM2274	SEEG9184_09070	SEN_RS11735	conserved	unnamed	SEEG pseudogene	MFS transporter	transporter
STM2691	SEEG9184_07115	SEN_RS13585	conserved	unnamed	SEEG pseudogene	type I secretion system permease/ATPase: TolC family OMP	virulence factor
STM3658	SEEG9184_00930	SEN_RS18105	conserved	*yiaH*	SEEG pseudogene	acetyltransferase	biosynthesis
STM1054	SEEG9184_09275	SEN_RS23040	conserved	unnamed	SEN pseudogene, SEEG 78 bp tRNA region	Gifsy-2 prophage protein in STM: GC rich region has a 7 bp deletion in SEN in a guanidine rich fragment, causing a frameshift; homology in SEEG to tRNA-pro	phage associated
STM4039	SEEG9184_23265	SEN_RS24515	conserved	unnamed	SEN, SEEG pseudogene	HO protein	inner membrane protein
STM1666	no homolog	SEN_RS07090	conserved	unnamed	STM pseudogene, SEEG absent	STM has in-frame stop following codon 24; SEN, hypothetical protein	unknown
STM1550	no homolog	SEN_RS07790	Deletion of 154 bp in SEN	unnamed	SEN pseudogene, SEEG absent	type II toxin-antitoxin system mRNA interferase toxin	cellular detoxification
STM0341	SEEG9184_19090&19095	SEN_RS01660	SEEG 1 bp deletion	unnamed	SEEG pseudogene or split into two genes	STM and SEN, putative inner membrane protein; SEEG, 2 transmembrane regulators	inner membrane protein
STM1130	SEEG9184_15700	SEN_RS05135	SEEG 1 bp deletion	unnamed	SEEG pseudogene	N-acetylneuraminic acid mutarotase	metabolism
STM1602	SEEG9184_13085	SEN_RS07535	SEEG 1 bp deletion	*sifB*	SEEG pseudogene	effector protein SifB	virulence factor
STM1698	SEEG9184_13605	SEN_RS06925	SEEG 1 bp deletion	*steC*	SEEG pseudogene	secreted effector kinase SteC	virulence factor
STM1869	SEEG9184_14565	SEN_RS06020	SEEG 1 bp deletion	unnamed	SEEG pseudogene	HO protein	phage associated
STM1939	SEEG9184_15200	SEN_RS05535	SEEG 1 bp deletion	unnamed	SEEG pseudogene	putative glucose-6-phosphate dehydrogenase	metabolism
STM2129	SEEG9184_09800	SEN_RS11065	SEEG 1 bp deletion	*yegB*	SEEG pseudogene	multidrug transporter subunit MdtD	transporter
no homolog	SEEG9184_21510	SEN_RS22085	SEEG 1 bp substitution	*sefD*	SEEG pseudogene	adhesin and master global regulator of phase transition; often mutated in SEN due to 1bp del in adenine homopolymer 8mer	regulon, adhesion
STM1941	SEEG9184_15210	SEN_RS05525	SEEG, SEN 1 bp deletion	unnamed	SEN, SEEG pseudogene	HO protein	inner membrane protein
STM3674	SEEG9184_00845	SEN_RS18190	SEEG, SEN 1 bp substitution	*lyxK*	SEEG pseudogene	carbohydrate kinase	metabolism

^1^ Reference genomes are S. Typhimurium NC_003197.2 (STM), S. Enteritidis NC_011294.1 (SEN), and S. Gallinarum CP019035.1 (SEEG).

**Table 3 microorganisms-09-02075-t003:** AT 8+mer motif variation in 3 serotypes of *Salmonella enterica* subspecies I.

Serovar	Typhimurium	Enteritidis	Gallinarum	Other Information		
Accession	NC_003277.2	NZ_CP063701.1	CM001154.1			
Other Name	pSLT	pSENV	str. SG9			
Atmer Size	Description of Loci	Variation from *pSLT*	Variation from *pSLT*	*pSLT* Start	*pSLT* End	Gene Function
9mer	Intergenic(PSLT039-PSLT038)	1nt deletion	1nt deletion	27961	27969	*spvB-spvC*;58nt upstream from spvC start
10mer	Intergenic (PSLT041-PSLT042)	1nt deletion	2nt deletion	32324	32333	*spvR*-PSLT041: 10nt upstream from *spvR* start
9mer	PSLT076	present	absent	62299	62307	*traY:* conjugative transfer: oriT nicking
8mer	PSLT088	absent	present	69093	69100	*traC*: conjugative transfer: assembly
8mer	PSLT102	absent	absent	82835	82842	*traS*: conjugative transfer: surface exclusion
8mer	PSLT111	truncated	truncated	93512	93519	*finO*: conjugative transfer: regulation

## Data Availability

The database analyzed for this project can be found at the National Center for Biotechnology Institute (NCBI) at https://www.ncbi.nlm.nih.gov (accessed on 26 August 2021).

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
