# Peer review of "AT Homopolymer Strings in Salmonella enterica Subspecies I Contribute to Speciation and Serovar Diversity"

_microorganisms, 2021, doi:10.3390/microorganisms9102075_

Round 1

Reviewer 1 Report

The authors speculate that the content of homopolymers in Salmonella genomes contributes to the success of specific serotypes as foodborne pathogens.

The theory is interesting and worth exploring, but the data presented to support the hypothesis are superficial. The manuscript needs to be improved before it can be accepted for publication.

Specific points:

  1. Orsi et. al. examined the role of homopolymers and their putative role as regulatory sequences in bacteria. I would suggest the authors to look into this paper and include the evidence provided in their speculations. >> Orsi, R.H., Bowen, B.M. & Wiedmann, M. Homopolymeric tracts represent a general regulatory mechanism in prokaryotes. BMC Genomics 11, 102 (2010). https://doi.org/10.1186/1471-2164-11-102
  2. Methods are not clearly described and must be improved - e.g. please explain how number of expected homopolymers in each species was calculated. Where necessary include the formulas used.
  3. Table 2 provides important data that is however poorly presented and difficult to understand. Please include a summary figure that show e.g. the function/classes of genes/proteins where the differences were found in specific genomes. Importantly, interesting observations should be pointed out in the text. E.g. the homopolymers in prophage regions - describe and discuss in Discussion.
  4. Please provide information about the positions of the disrupted homopolymers within the gene/proximity of the gene - terminal positions would suggest regulatory functions
  5. The authors included other genera in their analysis but in discussion these results were not mentioned, please amend.

Author Response

RESPONSES TO REVIEWER 1

NOTE: A file showing comparisons of the original to the revision is provided because of the numerous changes throughout the revision. 

The authors speculate that the content of homopolymers in Salmonella genomes

contributes to the success of specific serotypes as foodborne pathogens.

The theory is interesting and worth exploring, but the data presented to support the

hypothesis are superficial. The manuscript needs to be improved before it can be

accepted for publication.

Specific points:

  1. Orsi et. al. examined the role of homopolymers and their putative role as regulatory

sequences in bacteria. I would suggest the authors to look into this paper and include

the evidence provided in their speculations. >> Orsi, R.H., Bowen, B.M. & Wiedmann,

  1. Homopolymeric tracts represent a general regulatory mechanism in prokaryotes.

BMC Genomics 11, 102 (2010). https://doi.org/10.1186/1471-2164-11-102

This reference has now been incorporated as a seminal reference throughout the manuscript. I don’t know how it got missed considering search terms used to develop the reference list. It is referenced at least 6 times in the revision.

  1. Methods are not clearly described and must be improved - e.g. please explain how

number of expected homopolymers in each species was calculated. Where necessary

include the formulas used.    

No specific formulas were used that are not accessed through the software listed in the manuscript. However more details were included about how the data were tabulated. Please see the file titled “3 Compare Materials and Methods”.  

  1. Table 2 provides important data that is however poorly presented and difficult to

understand. Please include a summary figure that show e.g. the function/classes of

genes/proteins where the differences were found in specific genomes. Importantly,

interesting observations should be pointed out in the text. E.g. the homopolymers in

prophage regions - describe and discuss in Discussion.

To address this issue, the original Table 2 is now Table S3. From Table S3, a subset of genes were extracted and put into the new Table 2 in order to focus on the 22 genes that were mutated and differentiated the 3 serovars Typhimurium, Enteritidis, and Gallinarum. Specific groups of genes were discussed further, those that had some connection with antimicrobials, especially transporters. Not much information was gleaned from phage-associated homopolymers, it is possible we will need some powerful algorithms to compare phage types. Serovar Enteritidis had a broadly used phage typing scheme until about 2000, and it would make an excellent subject for such a study. However, we do not have those algorithms. This will be discussed with our computational biologists.

  1. Please provide information about the positions of the disrupted homopolymers within

the gene/proximity of the gene - terminal positions would suggest regulatory functions  

Additional analysis was done and the results were included (Lines 296-303 in the revision): The location of the first adenine or thymine in homopolymer strings in the subset of mutated genes was located relative to start codons ATG or GTG. Results were that 8, 7, 6, and 1 gene(s) fell into the 1st, 2nd, 3rd, and 4th quartile of gene lengths. These findings thus agree in concept with others that homopolymer tracts tend to be found closer to the 5’ end of open reading frames [6]. However, we suggest that these data are more accurately summarized as AT 8+mers of S. enterica do not often occur near the 3’ end of genes. An area of future research is to determine if taxonomic classes of bacteria differ in location of mutable homopolymer strings within open reading frames.

  1. The authors included other genera in their analysis but in discussion these results

were not mentioned, please amend.

Specific text about how S. enterica compared to different Bacteria is now included (lines 210 – 228): Results of comparisons between Bacteria are shown in Table 1. Details include: 1) AT 8+mers in S. enterica groups were significantly more frequent than what was observed for E. coli (p < 0.005); 2) across bacterial classes Vibrio vulnificus cII had a minimum of 90.0 AT 8+mers and Proteus mirabilis had a 712.7; 3) standard deviations between strains in each Genera ranged from 2.3 for Yersinia pseudotuberculosis to 84.1 for Enterococcus faecalis; 4) all the genera examined, including those that differed by Phylum and Class, had a relative paucity of GC 8+mers as compared to AT 8+mers; thus, it appears there is a bias for Bacteria maintaining AT 8+mers in genomes, or inversely, selecting against GC 8+mers; 5) each species appeared distinctly different from others; 6) Vibrio vulnificus had 180 and 90 AT 8+mers in chromosomes cI and cII respectively; thus, AT 8+mer content might be a chromosomal characteristic that maintains the separation of the two chromosomes.

Results based upon taxonomic class were an average of 379.57 and 430.8 AT 8+mers, respectively, for the Gammaproteobacteria and Bacilli. In addition, standard deviations were large for both classes, namely 174.51 and 251.24 respectively. The range reflected by standard deviations masks the individuality of each genus and species. For example, Bacillus anthracis had an average of 432 AT 8+mers, whereas Bacillus cereus had an average of 700.3. Results from S. enterica subspecies I had a relatively small overall standard deviation of 12.83 AT 8+mers across 24 serovars, which agrees with the classification of S. enterica subspecies I as a single genomic grouping.

Reviewer 2 Report

The manuscript of Guard et al. is presenting a hypothesis. The hypothesis is based on analysis of already existing complete genomes of Salmonella enterica serovars, E. coli and some other bacterial species. They are focusing on motifs of adenine and thymine homopolymers of at least 8 nucleotides in length. The entire manuscript is full of theories which still have to be demonstrated experimentally. The hypothesis is interesting and if this is in line with a concept of this journal to publish hypothesis, I am supporting it for a publication. So far, the instructions for authors are not predicting a form “hypothesis”. Anyway, I believe that the manuscript is not enough carefully expressing the assumption coming from their bioinformatic analysis, such as L257 (do we have experimental prove of that?). After reading the title, one would expect that the statement is supported with experimental proof, so the title has to be corrected and should reflect the fact that this is only at the stage of hypothesis.

Additional remarks:

  • Eubacteria: this is an old name for a domain Bacteria
  • Table 1: since domain is composed of millions of different species, here should be mentioned that representatives of some phyla (mention which) have been analyzed.
  • L106: reading this line, one would think that this is a number of genomes included in the analysis, but Table 1 reveal very low number of genomes which have been taken into the analysis. The number of genomes on which the theory of this manuscript is based is pretty low. If there are not more complete genomes available, the authors should sequence additional genomes by themselves.

Author Response

RESPONSES TO REVIEWER 2:

NOTE: A file showing comparisons of the original to the revision is provided because of the numerous changes throughout the revision.

P0 WORDS

The manuscript of Guard et al. is presenting a hypothesis. The hypothesis is based on analysis of already existing complete genomes of Salmonella enterica serovars, E. coli and some other bacterial species. They are focusing on motifs of adenine and thymine homopolymers of at least 8 nucleotides in length. The entire manuscript is full of theories which still have to be demonstrated experimentally. The hypothesis is interesting and if this is in line with a concept of this journal to publish hypothesis, I am supporting it for a publication. So far, the instructions for authors are not predicting a form “hypothesis”.

Thank you for understanding that we too believe the manuscript is best presented as hypothesis. The preliminary work generated has been used to procure funding for the next 5 years through USDA as part of a larger project, and thus we hope to generate experimental proof of the hypothesis over time.

Anyway, I believe that the manuscript is not enough carefully expressing the assumption coming from their bioinformatic analysis, such as L257 (do we have experimental prove of that?).

L257 reads “The AT 8+mer motif was located in genes and regulatory regions that impact phenotype, growth potential, virulence and metabolism of  Salmonella enterica subspecies I”. Yes, there is experimental proof and it was first described in the introduction. I have inserted references to that work, which generated the idea to review S. enterica AT homopolymers, at the end of the sentence.

That section, which starts the discussion, now reads (Lines 325-338 in the revision):

The AT 8+mer motif was shown here to be associated with 1) S. enterica speciation, 2) serovar designation via details of the AT 8+mer in plasmids, 3) hypermutability in genes and regulatory regions that impact phenotype, growth potential, virulence and metabolism of Salmonella enterica subspecies I. AT 8+mers influence microbial biology. For example, A and T homopolymers impact transcription termination in Archea [35]. The canine herpesvirus thymidine kinase gene has mutational hotspots at stretches of 8 adenines [36]. T7 bacteriophage RNA polymerases undergo transcription slippage at A and T homopolymers [37]. As mentioned previously for S. enterica serovar Enteritidis, a mutational hotspot in 1 of 8 adenines increased virulence [3]. We were not able to confirm that mutation in the motif was primarily in the 5’ region of genes; however, it is a subject that requires development of algorithms to address in detail [6]. These results suggest that the AT 8+mer motif facilitates maintenance of S. enterica as a persistent and important foodborne pathogen, as well as impacting epidemiological outcomes by contributing to ecological adaptation [38].

After reading the title, one would expect that the statement is supported with experimental proof, so the title has to be corrected and should reflect the fact that this is only at the stage of hypothesis.

            The title was corrected.

Additional remarks:

Eubacteria: this is an old name for a domain Bacteria

            Eubacteria has been changed to Bacteria throughout.

Table 1: since domain is composed of millions of different species, here should be mentioned that representatives of some phyla (mention which) have been analyzed.

            The table was edited as requested, and phyla are listed in the first column now.

L106: reading this line, one would think that this is a number of genomes included in the analysis, but Table 1 reveal very low number of genomes which have been taken into the analysis. The number of genomes on which the theory of this manuscript is based is pretty low. If there are not more complete genomes available, the authors should sequence additional genomes by themselves.

The database at NCBI is exploding so much that redundancy of effort has crept in. I am including new text in Materials and Methods to alert the reader that NCBI has a project in place to make sure quality of sequence data is maintained. The new text is (Lines 124 – 131): For S. enterica subspecies I, 12 complete genomes were assessed for Typhimurium, Enteritidis, and Typhi, and another grouping included a mixture of 12 serovars often recovered from foods. It is important to note that the NCBI is in the process of consolidating redundancy in over-represented genomes, thus the total number of genomes represented by high quality and well-annotated accession numbers may be larger than it appears as redundant material is condensed and re-annotated (https://www.ncbi.nlm.nih.gov/refseq/about/prokaryotes/reannotation/ ).

In response to sequencing genomes by ourselves, we added 91 genomes to the database in 2020, including 1 new annotated reference genome (DOI: 10.1016/j.ygeno.2019.04.005). These were the basis for finding that AT 8+mers were interesting (reference 3). At this time the Food Safety Inspection Service, the Food and Drug Administration, and the Centers for Disease Control are the agencies with funding for major sequencing projects. Even they have not been able to get through a back log of data to progress incompletely sequenced genomes to completed status. We are very much reliant on these organizations. We are very much hoping that the reannotation project will allow more efficient application of sophisticated algorithms for analysis of genomic motifs such as ours and others.

Reviewer 3 Report

Tabe 2 in its current format will be very hard to print. It should be split over 2 (maybe 3) pages, preferably in landsacape orieantation.

The contributions of the last three authors ( Michael J Rothrock Jr, Adelumola Oladeinde  and Devendra H Shah) are not defined, while they should be.

Author Response

RESPONSES TO REVIEWER 3”

NOTE: A file showing comparisons of the original to the revision is provided because of the numerous changes throughout the revision. 

Table 2 in its current format will be very hard to print. It should be split over 2 (maybe 3) pages, preferably in landscape orientation.

This was a great critique that resulted in turning the original Table 2 into Table S3. From Table S3 a subset of 22 genes that were of most interest due to mutations was extracted and formatted to be the new Table 2, which is much easier to access by readers rather than being a massive list. Another reviewer had a similar comment.

The contributions of the last three authors ( Michael J Rothrock Jr, Adelumola Oladeinde and Devendra H Shah) are not defined, while they should be.

            The roles of these 3 authors was defined better. Thank you for catching that omission.

Round 2

Reviewer 1 Report

The authors responded satisfactorily to the questions and followed majority of suggestions given. Moreover, substantial changes to the text were made that improved the readability and clarity of the observations described.

One last suggestion related to the new paragraph 3.6 should however be addressed. The association of homopolymers and the serovars of Salmonella based on their content in mobile elements should be made with extreme caution. Considering the fact that Salmonella's plasmids have often been acquired from other bacterial genera within Enterobactericeae family and contain large mosaic integrons and transposons that have been mixed and matched over the years, how can the authors make an assumption that these elements can be serovar specific? That would suggest that the plasmids co-evolved with the serovar populations, which often is not completely the case as plasmids are lost, acquired and modified all the time. Moreover the plasmids found in different serovars vary substantially and a direct comparison of plasmids with high genomic identities is not possible (as they don't exist).  I would suggest that the authors change this paragraph to describe variation of the motif in specific plasmids without associating this observation with serovar specificity .

Of note, Salmonella can have carry much bigger plasmids than the ones described e.g. IncF plasmids can be more than 200 kb.

Author Response

RESPONSE

Thank you for your comments. I edited this section to make it a more cautious presentation of serovar specificity of plasmids. The abstract was also addressed. I went back over the literature, and I continue to be impressed by studies that show serovar specific patterns of plasmids. Studies of serovar Enteritidis have even concluded that patterns are not distinctive enough to be used for sub-classification of strains though they can be used to identify Enteritidis. However, your point is well-taken that it is better to make a cautious statement at this time and indicate it is an area requiring further research. I am including a file showing the changes made in the abstract and the plasmid section to address your concerns. Other than this section, the rest of the manuscript is unchanged.

Reviewer 2 Report

The manuscript has been corrected accordingly. 

Author Response

Thank you so much for your review, it helped make this manuscript much better. As far as I can tell, your critiques were answered satisfactorily.